# Hair Follicle Classification and Hair Loss Severity Estimation Using Mask R-CNN

**DOI:** 10.3390/jimaging8100283

**Published:** 2022-10-14

**Authors:** Jong-Hwan Kim, Segi Kwon, Jirui Fu, Joon-Hyuk Park

**Affiliations:** 1Mechanical & Systems Engineering, Korea Military Academy, Seoul 01805, Korea; 2J-Solution, Daegu 41566, Korea; 3Mechanical and Aerospace Engineering, University of Central Florida, Orlando, FL 32816, USA

**Keywords:** Mask R-CNN, hair loss, hair follicle detection, image segmentation, and classification, hair severity estimation

## Abstract

Early and accurate detection of scalp hair loss is imperative to provide timely and effective treatment plans to halt further progression and save medical costs. Many techniques have been developed leveraging deep learning to automate the hair loss detection process. However, the accuracy and robustness of assessing hair loss severity still remain a challenge and barrier for transitioning such a technique into practice. The presented work proposes an efficient and accurate algorithm to classify hair follicles and estimate hair loss severity, which was implemented and validated using a multitask deep learning method via a Mask R-CNN framework. A microscopic image of the scalp was resized, augmented, then processed through pre-trained ResNet models for feature extraction. The key features considered in this study concerning hair loss severity include the number of hair follicles, the thickness of the hair, and the number of hairs in each hair follicle. Based on these key features, labeling of hair follicles (*healthy, normal, and severe*) were performed on the images collected from 10 men in varying stages of hair loss. More specifically, Mask R-CNN was applied for instance segmentation of the hair follicle region and to classify the hair follicle state into three categories, following the labeling convention (*healthy, normal and severe*). Based on the state of each hair follicle captured from a single image, an estimation of hair loss severity was determined for that particular region of the scalp, namely *local hair loss severity index (P)*, and by combining *P* of multiple images taken and processed from different parts of the scalp, we constructed the *hair loss severity estimation (*$P_{avg}$*)* and visualized in a heatmap to illustrate the overall hair loss type and condition. The proposed hair follicle classification and hair loss severity estimation using Mask R-CNN demonstrated a more efficient and accurate algorithm compared to other methods previously used, enhancing the classification accuracy by 4 to 15%. This performance supports its potential for use in clinical settings to enhance the accuracy and efficiency of current hair loss diagnosis and prognosis techniques.

## 1. Introduction

*Androgenetic alopecia*, or hereditary hair loss, is the most common form of hair loss in men and women [1], affecting up to 50 percent of the population worldwide [2,3]. *Telogen effluvium* is considered the second most common form of hair loss [4], a change in the number of hair follicles with growing hair, and is associated with a thinning of hair on the scalp, more prevalent at the top of the scalp [5]. There are other types of hair loss such as *Alopecia areata* [1], *Anagen effluvium* [6], or *Traumatic alopecia* [7]. Hair loss is most commonly caused by genetics, hormonal changes, medical conditions such as diabetes and lupus, medications and supplements, or a normal part of aging [8]. The common hair loss diagnostic methods widely adopted and practiced in clinical settings are the hair pull test, the pluck test, a scalp biopsy, daily hair counts, or a trichoscopy [9]. Trichoscopy is currently considered the most effective method for diagnosis and prognosis of hair loss since it offers a means to diagnose and monitor the progression of hair loss by objectively evaluating hair condition, hair follicles, and the scalp, including the total number of hairs on the scalp and the diameter of each strand of hair [10,11,12,13]. However, this method heavily relies on visual inspection of the digital images from Trichoscopy, is subject to human error, and leads to different results between clinicians. Additionally, there is a high cost and time associated with educating and training an individual to perform trichoscopy, which generally takes more than half a year and costs about 14,000 USD [14].

Due to these limitations, more advanced tools and techniques have been developed and tested that leverage computer-based processing and analysis, particularly deep learning. The multilayer feed-forward neural network, a type of Artificial Neural Network, trained using a back propagation algorithm, was proposed, which predicts hair loss based on the eight attributes: gender, age, genetic factors, surgery, pregnancy, Zinc deficiency, iron deficiency, anemia, and the use of cosmetics. This method showed high accuracy with a two-layer neural network used with the Levenberg-Marquardt algorithm [14]. Difference-of-Gaussian (DoG) filters were also considered for follicle detection using the Support Vector Machine (SVM) classifier [15]; however, this method depends on the handpicked features from the microscopic scalp hair image, which are prone to errors. An unsupervised hair segmentation and counting scheme was also proposed, showing 95.3% precision and 88.6% recall in classifying wavy and curly hair [16], but, similarly, its main drawback was manual parameter selection. Other classification methods have been introduced, e.g., scalp blotch detection and hair condition assessment using microscopic images [17], a deep learning method to classify patterned baldness and hair loss levels from facial images [18] or classifying four common scalp hair symptoms (dandruff, folliculitis, hair loss, and oily hair) [14]. These approaches have shown promising outcomes; however, a more accurate, efficient, and robust method is needed, that more comprehensively infers local and scalp-level hair loss severity, such as hair counting or measuring the diameter of each strand of hair.

More recently, hair loss features were extracted by combining a scalp image taken from the microscope using grid line selection and eigenvalue to determine the progression of hair loss [19]. Further, a support vector machine (SVM) and a k-nearest neighbor (KNN) were utilized to train a machine learning model to classify healthy and hair loss conditions [20]. The feasibility of automating the process of hair density assessment was tested by measuring the number of hairs from hair follicles and their type using deep learning-based object detection as well as other methods [21], such as VGG-16 [22], EfficientDet [23], YOLOv4 [24], and DetectoRS [25]. The common shortcomings of these approaches are prevalent false detections on the number of hairs, inaccuracy in detecting hair follicles, and lacking entire scalp-level hair loss severity estimation.

To address the aforementioned limitations of existing hair loss detection algorithms, we propose an intelligent system that utilizes multitask learning using Mask R-CNN to detect and localize hair follicles and measure hair loss severity. The key features considered in this study pertaining to hair loss include the number of hair follicles, the thickness of the hair, and the number of hairs in each hair follicle. First, we collect and label microscopic images with the following three classes: *healthy*, *normal,* and *severe*. Subsequently, by training Mask R-CNN, annotated hair follicles are learned with Region of Interest (ROI) pooling, and simultaneously, a regressor is trained using segmented information whose output ranges from zero to one, corresponding to hair loss severity. Finally, using location-translated images, the hair loss severity map is drawn, and the hair loss severity index is calculated. The key remarks and contributions of this work are:Mask R-CNN-based hair pore segmentation and classification;local hair loss severity estimation using PLS, which is bounded on 0 to 1;overall hair loss severity estimation and visual representation of hair loss using a heatmap.

The remainder of this article is organized as follows: the proposed method is introduced in Section 2, the training and evaluation methods are explained in Section 3, experimental results and discussions are provided in Section 4, and finally, we conclude this work in Section 5.

## 2. Methods

In this section, we describe the central framework and process designed and implemented in this work, starting from the input image to the system output, i.e., hair follicle segmentation, and status classification. First, object detection and image segmentation methods incorporated in this framework are explained, then the procedure within the network architecture of Mask R-CNN is detailed, and lastly, the labeling scheme and method are described.

### 2.1. Key Concepts

Convolutional neural network (CNN) is generally used in computer vision for classification as it is a translation equivariance network [26]. Thus, changes in the location of the input value do not affect the extracted features in classifying each object. In contrast, object detection not only classifies the object but also finds out where the object is located through training the model with pre-defined bounding boxes per each image of the whole dataset, and it does not require translation invariances [27,28]. On the other hand, image segmentation: representative examples of image segmentation can be broadly categorized as semantic (or object) segmentation or instance segmentation [29,30]. Semantic segmentation divides the same objects into the same area or color. On the contrary, instance segmentation detects, delineates, and localizes each distinct object of interest appearing in an image. Thus, instance segmentation is more suitable for the detection of hair follicles and the classification of different states of hair follicles.

### 2.2. System Architecture

The process flow of the system architecture is shown in Figure 1. First, after collecting and resizing the input images, the pre-trained ResNet [31] models were used as a feature extractor, and the extracted features were used for bounding box regression. After obtaining a feature map, Region of Interest (ROI) aligns to locate the relevant areas on the feature map. Then, the trained model localized and classified hair follicles according to each class and implemented the process of extracting feature factors. This was done by taking the average value after obtaining the results by classifying each class according to the condition of the follicles, i.e., the number of hairs per unit area. Next, after performing the aforementioned process, the distribution of the extracted characteristic factors was analyzed to determine three classes: *healthy*, *normal*, and *severe*. The score for each hair follicle is calculated and normalized to a range between 0 and 1. Based on the state of each hair follicle captured from a single image, an estimation of hair loss severity was implemented at twelve local points on the top of the head and then determined for that particular region of the scalp, namely the local hair loss severity index (*P*). By combining *P* of multiple images taken and processed from different parts of the scalp, we constructed the hair loss severity estimation ($P_{avg}$) and graphically visualized in a heatmap to illustrate the overall hair loss type and condition.

### 2.3. Mask R-CNN

Mask R-CNN is one of the object detection and localization algorithms that uses deep learning and image segmentation, which has gained popularity across various fields, such as medical (e.g., [32]) and industrial (e.g., [33]) applications. The Mask R-CNN employed in this study is an extension of Faster R-CNN [34], which consists of two stages. The first stage checks the image, then generates the proposal, and the second stage classifies the proposal and produces the masks and bounding boxes. In addition, it performs the region proposal of an object through box regression and classification and classifies which class the image corresponds to. Mask R-CNN’s backbone network adopts ResNet-50 and ResNet-101. ResNet-50 operates on 25.6 million parameters with 50 network layers, and has the advantages of faster training speed and algorithm execution than ResNet-101. On the other hand, ResNet-101 operates on 44.5 million parameters with 101 network layers, thus, the training and algorithm execution speeds are comparably slower than ResNet-50 in exchange for enhanced accuracy.

### 2.4. Dataset

For the diagnosis of hair loss in the respective areas of the scalp, RGB images were collected by Dino-Lite Microscope which has the advantages of adjustable magnification, high-definition microscopic images, and a convenient device-PC interface via USB. A 200× magnification was used to take the scalp images. The real-time image acquisition from the microscope was achieved using OpenCV and Python, which allowed the storage of an image on a PC for data construction. Since hair loss in women is generally less severe than in men, we collected images only from men (all Asian) across ages from different locations of the scalp. Furthermore, at the time of data acquisition, hair loss on the side of the head and sideburns were not prevalent, thus the data were obtained primarily from the upper part of the scalp. After obtaining the official consent of the participants, a total of 600 images were collected from 10 men (between the ages of 25 and 55), and data labeling on each image was performed by clinicians, resulting in 10 to 45 region annotations per image.

The criteria for performing labeling were determined by three factors: hair follicle size, the number of hairs, and hair thickness. The label *severe* denotes the cases when the follicle radius is ten pixels or less, the number of hairs in the follicle is one, and the thickness of the hair is three pixels or less. The label *normal* denotes the cases when the follicle radius is greater than ten and less than twenty pixels, the number of hairs in the follicle is one, and the thickness of the hair is greater than three and less than six pixels. The label *healthy* denotes the cases when the follicle radius is greater than twenty pixels, the number of hairs in the follicle is two or more, and the thickness of the hair is greater than six pixels. In addition, since this training model is focused on hair loss severity, the learning was conducted without taking other factors into consideration such as red spots, oiliness, and dead skin cells. Based on the criteria, the total number of labeling was 24,012, among which 3836 were labeled as severe, 11,262 were labeled as normal, and 8914 were labeled as healthy in Table 1. To address the data imbalance between classes during training, a resampling technique was applied to under-sample Normal and over-sample Severe classes.

The training-to-test ratio of the dataset was set at 0.75 to 0.25. The VGG Image Annotation (VIA) tool was used for mask annotation—a circular boundary around each hair follicle detected—and accurate labeling performed on each hair follicle was validated by multiple clinicians in Figure 2.

## 3. Training and Evaluation

The training was conducted using a TensorFlow v2 framework running on Nvidia Titan x3 and a AMD Ryzen 9 3950X 16-Core processor with 32GB RAM in Ubuntu 18.04 5 LTS environment. In addition, the batch size between the process was 25, learning rate of 1 × 10^−4^, using Adaptive Moment Estimation (ADAM) as an optimizer. The training of the model was performed by fine-tuning the ResNet models pre-trained on the coco dataset. After training 50 epochs for the head and 50 epochs for the rest of the full body, the bbox loss converged and the training was terminated at that point. While 450 images were used for the training (75% of the data), 150 images were used for the test (25% of the data). Additionally, the proposed method performed instance segmentation which allows the detection and classification of the health condition of the hair follicle. It is worth noting that the training was performed with randomly selected images from the entire dataset without considering the regional context, i.e., which part of the scalp the image was taken from.

### 3.1. Training Schemes

Data augmentation was performed to improve the accuracy and quality of the test images used in training and testing the model. First, brightness and contrast were adjusted for consistency in the test images to compensate for the difference in the ambient light during image capture. Second, if the test image was flipped or slanted due to the camera angle, appropriate corrections were made to enhance the normality of the data. Lastly, Gaussian blur was used to improve the image recognition performance of the model. These steps improved the usability and accuracy of the images which, in turn, helped improve the performance and efficiency of the model training and testing. The bounding box of hair follicles was created based on the width and height of hair follicles Equations (1) and (2). Additionally, the number of hairs through follicle counting was used as one of the indicators to measure the hair loss severity, in addition to partial least squares (*PLS*) in Equation (3).
(1)$$L_{width}=x_{2}-x_{1},\;L_{height}=y_{2}-y_{1}$$
(2)$$Area_{ij}=L_{width}\times L_{height}$$
(3)$$PLS=\overset{}{\underset{1\leq i\leq m}{\sum }}\overset{}{\underset{1\leq j\leq n}{\sum }}\frac{{\hat{y}}_{ij}\times Area_{ij}}{Area_{ij}}$$
where *m*, *n* denotes the number of follicles and the number of images in the training dataset, respectively. $Area_{ij}$ is calculated as the area of bounding box per each follicle, and ${\hat{y}}_{ij}$ is the predicted label for each follicle. Thus, *PLS* is calculated as a weighted sum of each image considering the area of a bounding box.

The multitask loss function of Mask R-CNN combines the loss of classification, localization, and segmentation mask: *L = L**cls + L**box + L**mask*, where *L**cls* and *L**box* are the same as in Faster R-CNN [34]. The mask branch generates a mask of dimension *m* × *m* for each RoI and each class; *k* classes in total. Thus, the total output is of size km^2^. Due to the fact that the model learns a mask for each class, there is no competition between classes for generating masks. *L**mask* is defined as the average binary cross-entropy loss, only including *k*th mask if the region is associated with the ground truth class *k*. Detailed loss used in calculating each mask is shown in Equation (4) where $y_{ij}$ is the label of a cell (*i, j*) in the true mask for the region of size *m* × *m*, and ${\hat{y}}_{ij}^{k}$ is the predicted value of the same cell in the mask learned for the ground-true class *k*.
(4)$$L_{mask}=-\frac{1}{m^{2}}\overset{}{\underset{1\leq i,\;j\leq m}{\sum }}\left[y_{ij}log\left({\hat{y}}_{ij}^{k}+\left(1-y_{ij}\right)log\left(1-{\hat{y}}_{ij}^{k}\right)\;\right)\right]$$

### 3.2. Evaluation Schemes

The bounding box adaptively changes its size according to the follicle size, as shown in Figure 3. Since each image is slightly different in size, all images were processed after resizing them to have consistent 640 × 480 pixels, and the same scaling was applied to the annotation mask for training. Mask R-CNN was applied in the following process. First, after labeling the health condition of hair into three classes (healthy, normal, and severe), training was conducted based on the label information. Next, the hair loss severity was evaluated by the number of follicles and hair conditions through follicle segmentation. Since hair distribution and hair loss vary across different areas of the scalp, the data were acquired from twelve different locations on the scalp. At the model inference stage, each image took minutes to process and outputs the results, including the detection, localization, and classification of hair follicles, which is reasonably fast and efficient.

## 4. Results and Discussion

### 4.1. Hair Follicle Classification

An intelligent system that utilizes multitask learning using Mask R-CNN to detect and localize hair follicles and measure hair loss severity is proposed. For this, we collect and label microscopic images with the following three classes: *healthy*, *normal,* and *severe*. By training Mask R-CNN, annotated hair follicles are learned with ROI pooling, and a regressor is simultaneously trained using segmented information. The test results of the trained Mask R-CNN with ResNet model are described in Figure 3. In each image, bounding boxes with corresponding predicted labels are plotted with confidence probability. In addition, healthy to severe hair loss classes are shown respectively from Figure 3a–f.

Two models were utilized, ResNet-50 and ResNet-101, for hair follicle classification using Mask R-CNN. The performance of these two models was assessed for training and test datasets with four metrics commonly used in performance evaluation of deep learning models: precision, recall, f1-score, and accuracy. Overall, for all metrics evaluated for both training and test datasets, ResNet-101 outperformed ResNet-50 most likely due to having 2 times as many layers and 1.7 times as many parameters in ResNet-101 as compared to ResNet-50 (Appendix A).

Further, the performance of each model was assessed for each label individually for training and test datasets in Figure 4. The result confirmed the superior performance of ResNet-101 over ResNet-50 across all three labels for both training and test datasets. In the training dataset, ResNet-50 reached 66% to 74% whereas ResNet-101 reached 82% to 87%. In the test dataset, ResNet-50 reached 55% to 72% whereas ResNet-101 reached 76% to 86%. Comparably low performance was observed in both models for the severe label, 66% to 70% in ResNet-101 and 55% to 67% in ResNet-50, as compared to the other two labels, normal and healthy. This might have been due to smaller samples of severe (3836) than normal (11,262) and healthy (8914).

Furthermore, a confusion matrix was generated to compare the misclassification rate between the two models for both training and test datasets and normalized as shown in Figure 5. The misclassification percentages of the *severe* label for ResNet-50 training datasets showed 22.66% and 11.02% as *normal* and *healthy* labels, respectively, which are greater than that of the ResNet-101 training dataset, at 15.35% and 2.97%, respectively. Both models misclassified *severe* as *normal* more than twice than it did with *healthy*. Similarly, The misclassification percentages of the *normal* label for ResNet-50 training datasets showed 7.45% and 18.50% as *severe* and *healthy*, respectively, which are greater than that of the ResNet-101 training dataset, 4.31% and12.22%, respectively. Both models misclassified *normal* as *healthy* more than 1.5 times as they did with *severe.* Finally, The misclassification percentages of the *healthy* label for ResNet-50 training datasets showed 3.09% and 23.27% as *severe* and *normal*, respectively, which are greater than that of the ResNet-101 training dataset, at 0.00% and 15.79%, respectively. Both models misclassified *healthy* as *normal*, but not as much as they did with *severe*. Overall, the majority of misclassifications occurred between *severe* and *normal*, and between *normal* and *healthy*, as expected. The misclassification rate of ResNet-50 and RestNet-101 in training datsets were 0.274 and 0.165, respectively, supporting RestNet-101 performs better than ResNet-50. Similar trends were observed in the test datasets between ResNet-50 and RestNet-101, whose misclassification rates were 0.316 and 0.207, respectively. Notably, the misclassification rate of *severe* as *healthy* and vice versa was more than five times greater for ResNet-50 (0.071 in training and 0.114 in the test dataset) than ResNet-101 (0.010 in training and 0.025 in the test dataset). These results support the rationale of selecting ResNet-101 over ResNet-50 as a system architecture of Mask R-CNN.

To further validate the performance of the ResNet-101 model over other alternatives, we conducted a benchmark comparison with other models (EfficientDet and YOLOv4) reported in [21] which were similarly applied to hair follicle detection classification in Table 2. ResNet-50 showed better performance as compared to EfficientDet; however, it underperformed as compared to YOLOv4. On the other hand, ResNet-101 showed superior performance compared to all other models compared. Our method used a circular shape to mask the region of interest (ROI) to train the model, which is different from the rectangular ROI employed in [21]. This could have been the reason for the higher mAP values of ResNet-101 compared to other models because it more closely resembles the shape of actual hair follicles which might improve the Intersection over Union (IoU). More performance results between ResNet-50 and ResNet-101 are described in Table A1 of Appendix A.

### 4.2. Local Hair Loss Severity Index (P)

The outcomes of Mask R-CNN using ResNet-101 classifies all hair follicles detected as *healthy*, *normal,* or *severe*. Using this information, the hair loss severity index (*P*) was defined to represent the hair loss severity of the local-level image, normalized to range between 0 and 1 Equation (5).
(5)Pk=normalize(∑i=13nki(αi+β))
where *k* is the index assigned to a local-level image data; *n* is the number of hair follicles, and *i* is the label (*healthy*, *normal,* and *severe*); α is the weighing coefficient multiplied to *n*, [*healthy, normal, severe*] = [$3^{2},\;2^{2},1^{2}$], such that it signifies the contribution of *healthy* and *normal* hair follicles over *severe* hair follicles in the local hair loss severity index. The number of hair follicles (*n*) alone is an important factor contributing to hair density in a local area, thus accounted for in the equation using β.

### 4.3. Hair Loss Severity Estimation and Mapping

The local hair loss severity index expressing the hair loss severity of a section of the head offers a means to estimate the scalp-level hair loss state if multiple images are taken from different regions of the scalp and integrated. Thus, the top region of the scalp was divided into 12 sub-regions as shown in Figure 6, specifically the left, center, and right side of the scalp, each proportionately divided into four regions from the anterior to the posterior side. These 12 locations of the scalp were determined to capture the overall hair loss severity and type. The hair loss severity estimation, $P_{avg}$, was calculated as an average of the *P* values from 12 locations of the scalp Equation (6). $P_{avg}$ is a 0 to 1 scale that represents qualitatively an individual’s hair loss severity state. Here, *m* is the number of images which, in our case, 12.
(6)Pavg=1m(∑k=1mPk)

The mapping of the local hair loss severity index into a graphical representation of the scalp was carried out using a heatmap that visualizes the overall hair loss severity of the entire scalp. Figure 6 shows the three representative subjects’ data—overall healthy, moderate hair loss, and severe hair loss—to demonstrate the process from the local hair loss severity index calculation to hair loss severity estimation and mapping. First, representative images from the key regions of the scalp and their respective local hair loss severity index *P* are presented. Then, $P_{avg}$ was calculated for each case after processing Pk values of all 12 locations, which are in agreement with their hair loss severity: 0.879 for healthy, 0.565 for moderate hair loss, and 0.160 for severe hair loss. The hair loss severity mapping using a heatmap well depicts the regions of severe loss to healthy states as shown in Figure 6d–f. This offers an intuitive way to visualize the hair loss areas and conditions as well as different types of hair loss, such as type M, type U, or type O [35]. Further, such visual mapping of hair loss severity can be used as a diagnosis and/or prognosis tool, or to assess treatment outcomes if the mapping was done periodically.

### 4.4. Limitations and Future Work

The deep learning model architecture used in this proposed method is not a state-of-the-art method, hence the detection performance could be further improved with more advanced methods. Still, the comparison study conducted with some models frequently used in similar domains revealed that the ResNet-101 offers overall comparable or better performance.

Accurate detection of all hair follicles and constructing correct labeled boundaries from 2D images remain major challenges, especially when the hair follicles are covered by layers of hair (e.g., Figure 3c). This leads to an underestimation of the total hair follicles per unit area, which in turn can affect the accuracy of the local hair loss severity index. However, this situation is not so problematic in *normal* or *severe* hair loss as there is a lesser chance of obstruction of hair follicles by hair strands due to overall low hair density per unit area. Using multiple images taken from the same local area after combing hairs in different directions, processing them individually, and comparing their P values, could effectively alleviate this issue.

In terms of the overall hair loss severity estimation, $P_{avg}$, developed in this study, the accuracy of this estimation depends on the resolution of the scalp scanning. For example, if we increase the resolution from 12 regions to 24 or higher, this will yield slightly different results. It is a tradeoff between the time and cost associated with scalp scanning versus a possible improvement in the accuracy of hair loss severity estimation. To address this, a few case studies with varying hair loss states will be performed while adding intermediate imaging locations between those 12 points on the scalp.

The current equation to compute local hair loss severity index *P* accounts for the number of hair follicles for each of the three different follicle classifications (*healthy*, *normal,* and *severe*). Other factors that could be incorporated into the equation are the sum of the area of the hair follicles (bounding boxes) corresponding to each class. As shown in Figure 3, the size of hair follicles is different even within the same class. The summation of the area of all hair follicles would provide an indication of hair follicle density per unit area which can be an additional term to be included (5).

Since the dataset is collected from only Asian men with black hair, the generalizability of the dataset is limited to this specific population. Thus, some external factors which were not included in the dataset, such as different hair types (e.g, curly), colors (e.g., blond or gray), or skin tones, which should be further tested using the proposed algorithm to address its general applicability. Fine-tuning and optimization of the parameters in the network and the image processing routine should make it feasible to incorporate specificities of these external factors into the model. Once validated, broader applicability of the proposed method and obtaining correct inferences on hair loss severity could be achieved.

## 5. Conclusions

This paper presents an efficient and accurate way to detect and classify hair follicles and a method to estimate local scalp level and hair loss severity using a deep learning framework. Two ResNet model-based Mask R-CNNs were applied and the highest performance was recorded in 62.43%, 80.62%, 78.85%, and 79.29% in mAP, precision, recall, and accuracy, respectively. The key remarks and contribution of this work are (1) Mask R-CNN-based hair pore segmentation and classification, (2) local hair loss severity estimation using PLS, which is bounded on 0 to 1, (3) overall hair loss severity estimation and visual representation of hair loss using a heatmap.

The proposed algorithm and hair loss estimation schemes have clinical Implications as they can inform an individual’s hair loss state accurately and efficiently once scalp imaging is done. Thus, the system framework has the potential to be implemented in clinical settings, expanding and improving the practicality of automated and learning-based image processing and classification algorithms. As an extension to this paper, our future work will focus on validating the efficacy of this method on different hair curl types, hair/skin color, and other factors that have not been considered in this work.

## Figures and Tables

**Figure 1 jimaging-08-00283-f001:**
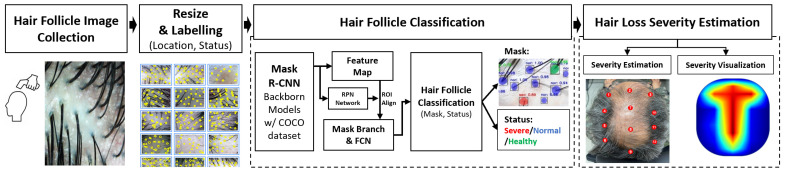
Model architecture of this paper.

**Figure 2 jimaging-08-00283-f002:**
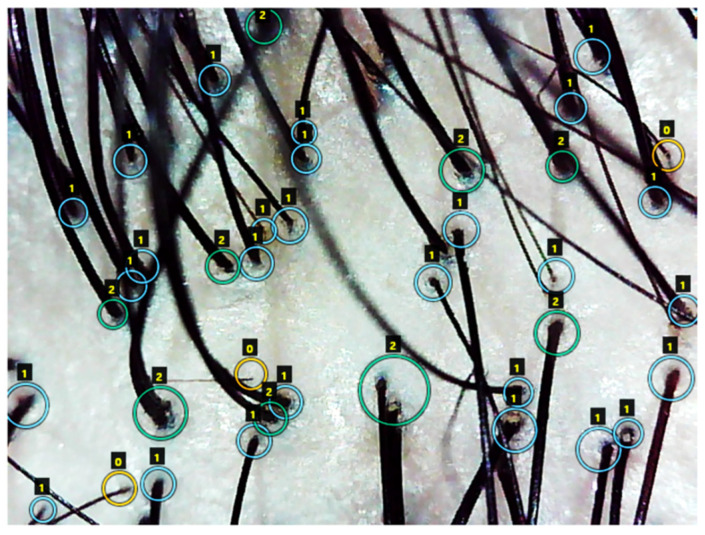
The label scheme of three classes using the VGG Image Annotation (VIA) tool; 0 and a yellow circle denotes “severe”, 1 and a blue circle denotes “normal”, and 2 and green circle denotes “healthy”.

**Figure 3 jimaging-08-00283-f003:**
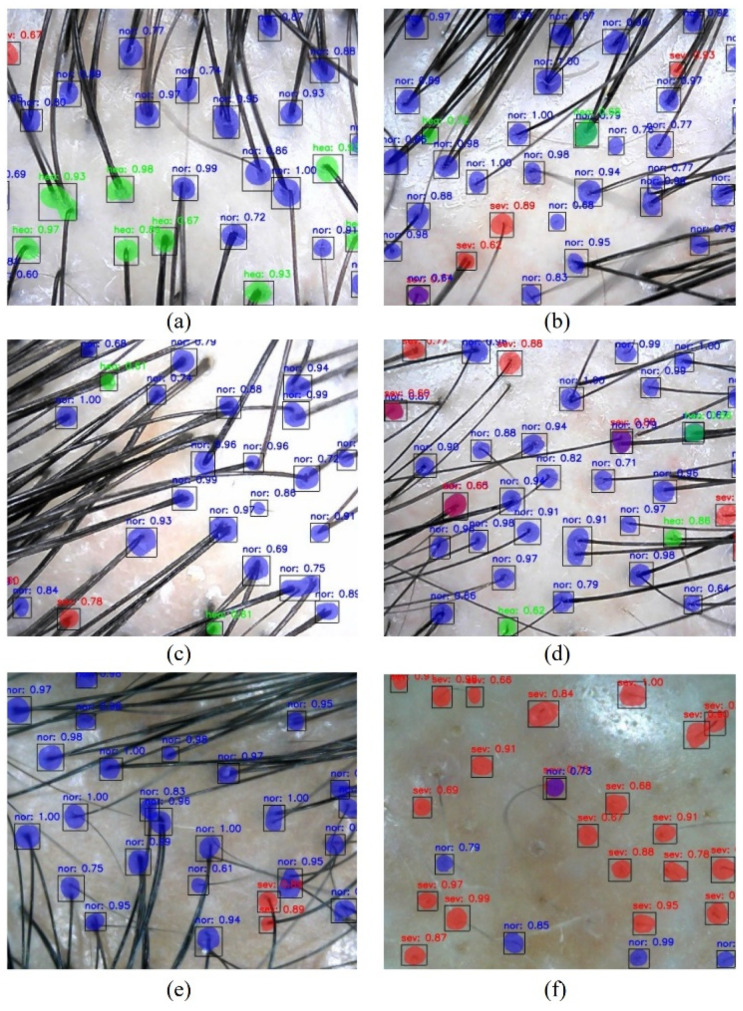
Test results of the trained Mask R-CNN with ResNet model as a feature extractor, where red circles indicate *severe* label, blue circles indicate *normal* label, and green circles indicate *healthy* label. Purple circles are due to different classifications of two very adjacent hair follicles, severe and normal. From healthy to severe hair loss classes are shown respectively from (**a**–**f**).

**Figure 4 jimaging-08-00283-f004:**
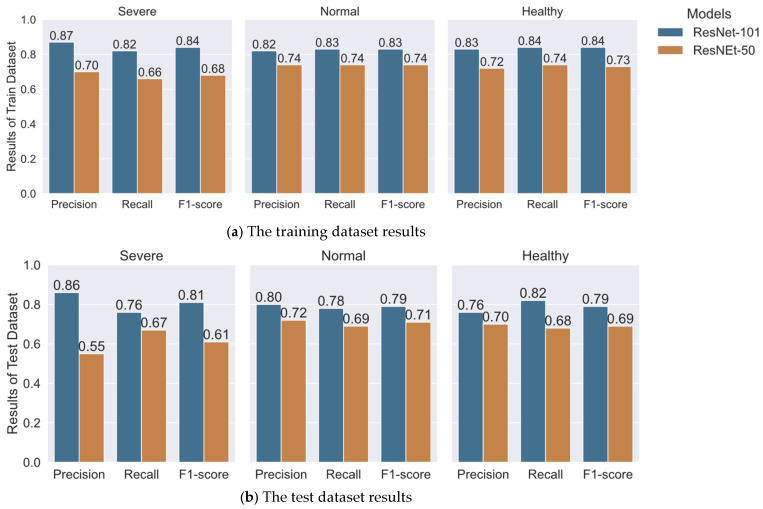
Hair follicle classification results of Train dataset in (**a**) and Test dataset in (**b**) compared between ResNet-101 and ResNet-50.

**Figure 5 jimaging-08-00283-f005:**
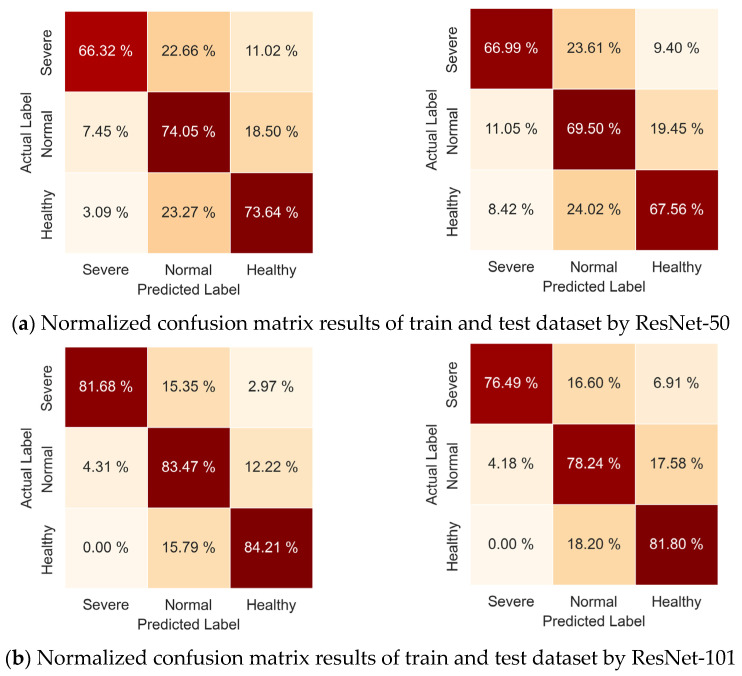
Confusion matrix results of hair follicle classification: ResNet-50 training dataset in (**a**)-left, ResNet-50 test dataset in (**a**)-right, ResNet-101 training dataset in (**b**)-left, and ResNet-101 test dataset in (**b**)-right.

**Figure 6 jimaging-08-00283-f006:**
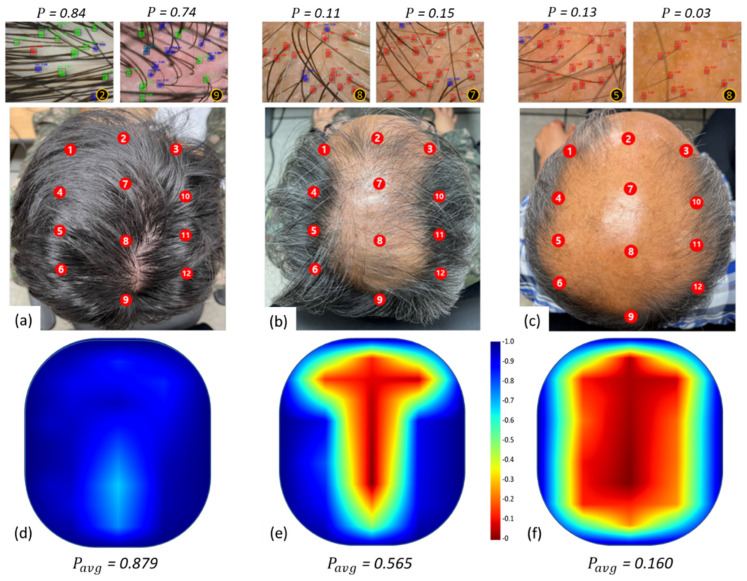
Representative top-view head photos and their respective hair loss severity heatmaps of the subject with the overall healthy state (**a**,**d**); moderate hair loss (**b**,**e**); and severe hair loss (**c**,**f**). On the top of the head photos, the local hair loss severity index P is presented with the images taken from different locations of the scalp. The hair loss severity estimation $P_{avg}$ for the healthy, moderate hair loss and severe hair loss is 0.879, 0565, and 0.160, respectively, reflecting a gradual decrease in the value as hair loss progresses.

**Table 1 jimaging-08-00283-t001:** Description of the dataset.

Dataset		Total	Train	Test
Images		600	450 (75%)	150 (25%)
Labels	Total	24,012	17,261	6751
	Severe	3836	2794	1042
	Normal	11,262	8059	3203
	Healthy	8914	6408	2506

**Table 2 jimaging-08-00283-t002:** Performance comparison between deep-learning models applied to hair follicle classification.

Models	mAP	mAP(50)	mAP(75)	Precision	Recall	Accuracy
EfficientDet [23]	31.97%	53.45%	35.38%	71.24%	64.09%	64.71%
YOLOv4 [23]	58.67%	73.11%	60.85%	80.75%	80.22%	75.73%
ResNet-50	54.21%	69.05%	56.23%	65.95%	68.02%	68.39%
ResNet-101	62.43%	77.36%	64.47%	80.62%	78.85%	79.29%

## Data Availability

Not applicable.

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
