# Peer review of "Hair Follicle Classification and Hair Loss Severity Estimation Using Mask R-CNN"

_2313-433X, 2022, doi:10.3390/jimaging8100283_

Round 1

Reviewer 1 Report

The authors presented a research study on hair follicle classification and hair loss severity estimation using a deep learning model. Three types of hair follicles are classified. The presented research study is scientifically sound. The section-wise comments: 

Abstract: 

I would suggest adding one or two sentences about the background of the study at the beginning of the abstract. The article will be able to summarize the whole work if the authors mention results in the abstract, for example, the accuracy of classification and hair loss severity estimation value. 

1. Introduction

The background of the research in the first four paragraphs is out of track. I would suggest shrinking the first four paragraphs into one paragraph. 

2. Methods

I would suggest explaining the block diagram in a separate paragraph instead of a caption. I don't think it is necessary to separate segmentation and classification blocks in 1. 

3. Training and Evaluation

Move figure 3 to the Result section

4. Reduce the size of the confusion matrix including the figure font size. No need to explain too much about the confusion matrix, the figure explains itself about it. Please convert absolute values in the confusion matrix into percentage values. 

5. conclusion: Better to add a sentence about the significance of this research, for example, why this research is important and contributes to a related area?  

Author Response

Comment #1

The authors presented a research study on hair follicle classification and hair loss severity estimation using a deep learning model. Three types of hair follicles are classified. The presented research study is scientifically sound. The section-wise comments:

Abstract:

I would suggest adding one or two sentences about the background of the study at the beginning of the abstract. The article will be able to summarize the whole work if the authors mention results in the abstract, for example, the accuracy of classification and hair loss severity estimation value.

  • Thank you for the suggestion. We’ve added a few sentences at the beginning of the abstract accordingly.
    • “Early and accurate detection of scalp hair loss is imperative to provide timely and effective treatment plans to halt further progression and save medical cost. Many techniques have been developed leveraging deep learning to automate the hair loss detection process. However, accuracy and robustness of assessing hair loss severity still remain as a challenge and barrier for transitioning such a technique into practice. The presented work proposes an efficient and accurate algorithm to classify hair follicles and estimate hair loss severity which was implemented and validated using a multitask deep learning method via Mask R-CNN framework.”
  • Also, as per your suggestion, we’ve added one sentence summarizing the result in the abstract:
    • “The proposed hair follicle classification and hair loss severity estimation using Mask R-CNN demonstrated more efficient and accurate algorithm compared to other methods previously used, enhancing the classification accuracy by 4 to 15%.”

  1. Introduction

The background of the research in the first four paragraphs is out of track. I would suggest shrinking the first four paragraphs into one paragraph.

  • Thanks for the suggestion. Now the first four paragraphs have been shrunk down to one paragraph by removing the details about hair loss conditions.

  1. Methods

I would suggest explaining the block diagram in a separate paragraph instead of a caption. I don't think it is necessary to separate segmentation and classification blocks in 1.

  • This is a good point. Thank you. We’ve revised accordingly by relocating the explanation in text, not in caption.
  1. Training and Evaluation

Move figure 3 to the Result section

  • It fits better in the Result section. Thank you.
  1. Reduce the size of the confusion matrix including the figure font size. No need to explain too much about the confusion matrix, the figure explains itself about it. Please convert absolute values in the confusion matrix into percentage values.
  • The size of the confusion matrix has been reduced. Absolute values have been converted to relative values
  1. conclusion: Better to add a sentence about the significance of this research, for example, why this research is important and contributes to a related area?
  • Thanks for the suggestion. We’ve added the following in the conclusion to highlight the importance and implication of the presented work.
    • “The proposed algorithm and hair loss estimation schemes have clinical implications as it can inform an individual’s hair loss state accurately and efficiently once scalp imaging is done. Thus, the system framework has the potential to be implemented in clinical settings, expanding and improving the practicality of automated and learning-based image processing and classification algorithms.”

Thank you so much for your time and valuable inputs. We greatly appreciate it.

Reviewer 2 Report

The paper presents a novel method of hair loss severity assessment based on the state of hair and hair follicles in microscopic images. The dataset includes scalp images of Asian men. The photos are first subjected to ResNet-based segmentation. Obtained hair/hair follicles are simultaneously labeled as characteristic for a healthy/normal/severe state. Results are aggregated, and several indexes are introduced to describe local and total hair loss severity.

The method is interesting and well-designed. However, several issues should be addressed or discussed:

  1. There are some recent papers mentioning 'ESENSEI Challenge: Marking Hair Follicles on Microscopic Images' (apparently https://knowledgepit.ai/post/311/), notably http://dx.doi.org/10.15439/2018F389, concerning a similar segmentation task and featuring similar F-scores. Please consider including this in the introduction/discussion.
  2. During the labeling (lines 190-onward), size criteria were used to label hair follicles as healthy/normal/severe. Fig.2 depicts the process. However, as it is suggested in line 240, during data augmentation, photos might have required resizing or slanting due to a camera angle. Was a similar operation performed before labeling?
  3. Could artificial lighting and a tripod or stand be used to position the microscopic camera in further work, increasing the quality of raw images? Please discuss.
  4. As given in Table 1, "Normal"-classified follicles are in the majority of all processed. The confusion matrix suggests that the performance is consistent across classes (e.g., ~70% for ResNet-50 labeling Test dataset). Was the proportion of all classes equalized somehow before training?
  5. On the other hand, the loss severity index (P) assigns different weights for healthy/normal/severe follicles (3^2, 2^2, 1^2). Apparently, performance in a healthy class is most important for the validity of P index. Please discuss whether you think this should be accounted for in training or left as is (assuming perfect labeling at some further point).

What is more:

1. is the y^_ij component in Eq. 3 the same as the 0/1/2 label in the earlier text?

2. The last sentence in Section 3 (line 235-) could be moved as to be the second to last.

3. Colors in Fig. 2 do not fully resemble the description (especially red).

4. ResNet models pre-trained on coco dataset are mentioned in line 230. Is it possible to provide references to that network?

5. Some spurious text is available in line 110 (First Bullet).

6. Line 133: the end of the sentence is missing.

Author Response

Comment #2

The paper presents a novel method of hair loss severity assessment based on the state of hair and hair follicles in microscopic images. The dataset includes scalp images of Asian men. The photos are first subjected to ResNet-based segmentation. Obtained hair/hair follicles are simultaneously labeled as characteristic for a healthy/normal/severe state. Results are aggregated, and several indexes are introduced to describe local and total hair loss severity.

The method is interesting and well-designed. However, several issues should be addressed or discussed:

  • There are some recent papers mentioning 'ESENSEI Challenge: Marking Hair Follicles on Microscopic Images' (apparently https://knowledgepit.ai/post/311/), notably http://dx.doi.org/10.15439/2018F389, concerning a similar segmentation task and featuring similar F-scores. Please consider including this in the introduction/discussion.
  • Thank you for the suggestion. We’ve added this paper in the introduction section where we explain previous works about deep learning based hair follicle detection. This paper only reports F-score, not Recall, Precision or Accuracy, so it limits comparison between our model and theirs. Our model seems to offer better result (5.6% higher F-score) compared to theirs. Still, this is very relevant paper thus we’ve added as a reference.
  • During the labeling (lines 190-onward), size criteria were used to label hair follicles as healthy/normal/severe. Fig.2 depicts the process. However, as it is suggested in line 240, during data augmentation, photos might have required resizing or slanting due to a camera angle. Was a similar operation performed before labeling?
  • Thanks for pointing this out. We used a microscope camera to capture the images which was mounted on a transparent support frame. There was not much of resizing or adjusting of images needed because the angle and distance of the camera to the head were consistent across participants. Although the images were taken with 200x microscope thus the effect of camera angle was minimal, when happen resizing and slanting of images due to a camera angle, appropriate correction were made to enhance the normality of data as described in the paper and similarly implemented before labeling

  • Could artificial lighting and a tripod or stand be used to position the microscopic camera in further work, increasing the quality of raw images? Please discuss.
  • Thanks for the comment and suggestion. Indeed, with controlled lighting and more stable apparatus for image capturing would increase the quality of raw images. We’d consider incorporating this idea in our future data collection.

  • As given in Table 1, "Normal"-classified follicles are in the majority of all processed. The confusion matrix suggests that the performance is consistent across classes (e.g., ~70% for ResNet-50 labeling Test dataset). Was the proportion of all classes equalized somehow before training?
  • Thank you for the comment. The dataset we used in this study has data imbalance issue; the number of images of the normal class are comparably higher than that of the other classes. This is an issue inherent from our dataset which we tackled with a “resample technique” – during training we undersampled the normal classes and oversampled the severe cases – which partially addressed the issue. Nevertheless, misclassifications were primarily from normal vs. severe, and normal vs. healthy, which show that resampling technique could not fully address the issue. We’ve added a section explaining this process in the data collection and discussion sections.
    • “To address the data imbalance between classes during training, a resampling technique was applied to under-sample Normal and over-sample Severe classes.”
  • On the other hand, the loss severity index (P) assigns different weights for healthy/normal/severe follicles (3^2, 2^2, 1^2). Apparently, performance in a healthy class is most important for the validity of P index. Please discuss whether you think this should be accounted for in training or left as is (assuming perfect labeling at some further point).
  • Thank you for the comment. When we calculate Hair Loss severity Index (P), the weight put on healthy is much higher (9) then normal (4) and severe (1). This is because we wanted to express P in [1,0] where 1 is healthy and 0 is severe, and as hair loss progresses the rate at which this index decrease will be exponential. That said, these weighing coefficients are set by the authors, and need to be tuned in future work to optimize the weights. Hair loss severity index (P) is computed from the outputs of the model, and is not associated with training/test results. The accuracy of P will be dictated by the accuracy of the classification.

What is more:

  1. is the y^_ij component in Eq. 3 the same as the 0/1/2 label in the earlier text?
  • Y^ij represents the label of 0, 1, and 2.

  1. The last sentence in Section 3 (line 235-) could be moved as to be the second to last.
  • Thank you for the suggestion. Relocated accordingly.

  1. Colors in Fig. 2 do not fully resemble the description (especially red).
  • “yellow circle denote “severe”, 1 and blue circle denote “normal”, and 2 and green circle denote “healthy”.

  1. ResNet models pre-trained on coco dataset are mentioned in line 230. Is it possible to provide references to that network?
  • Absolutely.  We’ve found the right reference and added.

  1. Some spurious text is available in line 110 (First Bullet).
  •  Corrected.

  1. Line 133: the end of the sentence is missing. 
    •   Corrected.

Thank you so much for your time and valuable inputs. We greatly appreciate it.
